# Incidence and Etiology of Postharvest Fungal Diseases Associated with Bulb Rot in Garlic (*Alllium sativum*) in Spain

**DOI:** 10.3390/foods10051063

**Published:** 2021-05-12

**Authors:** Laura Gálvez, Daniel Palmero

**Affiliations:** Department of Agricultural Production, School of Agricultural, Food and Biosystems Engineering, Universidad Politécnica de Madrid, Avda. Puerta de Hierro, 4, 28040 Madrid, Spain; laura.galvez@upm.es

**Keywords:** *Fusarium proliferatum*, *Penicillium allii*, identification, pathogenicity test, virulence

## Abstract

In recent years, different postharvest alterations have been detected in garlic. In many cases, the symptoms are not well defined, or the etiology is unknown, which further complicates the selection of bulbs during postharvest handling. To characterize the different symptoms of bulb rot caused by fungi, garlic bulb samples were collected from six Spanish provinces in two consecutive years. Eight different fungal species were identified. The most prevalent postharvest disease was Fusarium dry rot (56.1%), which was associated with six Fusarium species. *Fusarium proliferatum was* detected in more than 85% of symptomatic cloves, followed by *F. oxysporum* and *F. solani*. Pathogenicity tests did not show a significant correlation between virulence and mycotoxin production (fumonisins, beauvericin, and moniliformin) or the mycelial growth rate. *Penicillium allii* was detected in 12.2% of the samples; it was greatly influenced by the harvest season and garlic cultivar, and three different morphotypes were identified. *Stemphylium vesicarium* and *Embellisia allii* were pathogenic to wounded cloves. Some of the isolated fungal species produce highly toxic mycotoxins, which may have a negative impact on human health. This work is the first to determine the quantitative importance, pathogenicity, and virulence of the causative agents of postharvest garlic rot in Spain.

## 1. Introduction

Garlic is a crop of great importance throughout the world and is highly valued for its culinary and medicinal properties. Its secondary metabolites have been shown to positively affect health and contribute to the prevention of many common human diseases, particularly through their antioxidant, anti-inflammatory, and lipid-lowering effects [1].

Spain, the first producer of garlic in Europe, has a current harvested area of 27,350 ha [2]. In recent years, garlic producers have faced new sources of postharvest losses. Diseases, pests, and poorly characterized physiological disorders cause significant postharvest garlic losses, but discriminating the symptoms of these agents remains challenging.

Postharvest rot is one of the main causes of garlic bulb loss during storage. Its incidence is directly related to the presence of pathogens on harvested bulbs and is greatly influenced by postharvest handling processes, including drying, storage, transportation, and marketing of the bulbs [3,4]. However, many pathogens initiate their infection during the development of plants in the field [5]; and the disease then progresses during the postharvest period. Some microorganisms may remain dormant and then develop during the postharvest period, causing losses of economic importance [6] or acting as inoculum reservoirs in cloves used for sowing.

The most important fungal pathogens known to cause garlic bulb rots are *Sclerotium cepivorum*, *Penicillium* spp., and *Fusarium* spp., which cause white rot, green rot, and dry rot, respectively [7,8]. Other fungal diseases are caused by *Embelllisia allii*, *Botrytis porri*, and *Aspergillus niger*, which lead to skin blotch and black mold on garlic bulbs [9,10,11]. However, few studies have described the quantitative importance of these diseases. Dugan et al. [12] described the incidence of the fungal pathogens *Aspergillus niger*, *A. ochraceus*, *Botrytis porri*, *Embellisia allii*, *Fusarium oxysporum* f. sp. *cepae*, *F. proliferatum*, and *Penicillium hirsutum* in garlic bulbs in the United States. Valdez et al. [13] cited the importance of the green rot caused by *Penicillium allii* in Argentina. In Egypt, four fungal genera were detected in garlic bulbs, namely *Aspergillus*, *Botrytis*, *Fusarium*, and *Penicillium* [14], of which *Fusarium* spp. were the most prevalent. In a recent study conducted in Italy, *F. proliferatum* was the predominant fungus in bulbs infected during the postharvest period [15]. In Spain, *F. proliferatum* was previously described as one of the main garlic pathogens [6], but the quantitative importance of the disease that it causes in different garlic cultivars remains unknown, as does the incidence of the other fungal diseases in garlic.

Moreover, many of these fungi are known to produce a range of toxins, including *fumonisins*, *moniliformin*, *beauvericin*, *fusaproliferin*, and *fusaric acid* [16,17,18]. Most mycotoxins are chemically stable and survive food processing; therefore, they present a great risk to the health of humans and livestock [15].

In garlic dryers or the warehouses in which bulbs are selected, early detection of garlic damaged by fungi is essential to removing the disease-causing agent from the food chain. Fungal damage is usually not visible and therefore not detected during the first handling step, but postharvest diseases progress during transportation and storage, and they are the main reason that garlic bulbs are rejected by the international market. To implement successful control strategies, it is necessary to differentiate these symptoms and determine the etiology of different fungal diseases.

The objectives of this work were to (1) characterize the different types of clove rot caused by fungi and determine their quantitative importance, (2) identify the fungal microorganisms associated with different symptoms, and (3) determine the pathogenicity and virulence of isolated fungi toward non-wounded and wounded cloves.

## 2. Materials and Methods

### 2.1. Plant Samples

Garlic bulbs were collected in two consecutive harvest seasons (2013 and 2014) from different Spanish agricultural cooperatives, particularly from three provinces in the Castilla La-Mancha region (Cuenca, Albacete, and Toledo) and three provinces in the Castilla y León region (Zamora, Valladolid, and Ávila). Symptomatic bulbs of different types, including Purple Chinese (cv. Garpek), Purple Spanish (cv. Morado de Cuenca), White Spanish (cv. Garcua), and White Chinese (cv. Spring), were transported to the Plant Protection Lab (Universidad Politécnica de Madrid, Spain). In total, 1264 cloves of all types of garlic from twelve cooperatives were analyzed (Figure 1).

### 2.2. Sample Processing and Description of Symptoms

Garlic bulbs were peeled, and the cloves were separated for external visual analysis. Symptomatic cloves were then classified according to the following morphological characteristics of the lesion: color, surface appearance, contour, consistency, and the presence or absence of mycelia, as well as the color and consistency of the internal lesions.

### 2.3. Fungal Isolation

Symptomatic portions of flesh were cut from all garlic cloves that showed signs of disease. The samples were surface disinfected for 3 min in a 2% sodium hypochlorite solution and rinsed twice in sterile distilled water. Four small pieces cut from the margin between healthy and symptomatic tissue were plated on potato dextrose agar (PDA). The plates were incubated at 25 °C in the dark until fungal colonies were large enough to be examined and annotated. Initial fungal isolation was performed by transferring the growing mycelia to a Petri plate containing PDA, and monosporic cultures were obtained. Isolates of *P. allii* were incubated in the dark for 7 days and morphologically characterized on Czapek yeast agar (CYA), according to Valdez et al. [13].

### 2.4. Identification of Fungal Isolates

Preliminary identifications were performed on the basis of the colony characteristics and micromorphology using a microscope and taxonomic criteria [19,20,21,22,23,24,25,26]. Identification at the species level was confirmed using molecular methods recommended for each taxonomic group. After 7 days of incubation at 25 °C on PDA in Petri dishes, the mycelia were collected into Eppendorf tubes. DNA was extracted according to the protocol of [27]. The concentration of DNA was estimated by a NanoDrop ND-1000 spectrophotometer (NanoDrop Technologies). The ITS1- 5,8S-ITS2 (ITS) region of nuclear rDNA was amplified with the primers ITS1 (5′-TCCGTAGGTGAACCTGCGG-3′) and ITS4 (5′-TCCTCCGCTTATTGATATG-3′). PCR was performed using an initial cycle of denaturation at 95 °C for 4 min, 40 cycles of denaturation at 95 °C for 30 s, annealing at 50 °C for 30 s, extension at 72 °C for 1 min, and a final extension at 72 °C for 3 min. Amplification of the partial β-tubulin (*TUB*) gene was performed using the sequence primers Bt2a (5′-GGTAACCAAATCGGTGCTGCTTTC-3′) and Bt2b (5′-ACCCTCAGTGTAGTGACCCTGGC-3′) and the following PCR conditions: an initial cycle of denaturation at 94 °C for 1 min 25 s, 32 cycles of denaturation at 94 °C for 45 s, annealing at 58 °C for 55 s, extension at 72 °C for 30 s, and a final extension at 72 °C for 5 min. Amplification of the partial sequence of the translation elongation factor 1α (EF-1α) gene was performed using the primers EF1T (5′- ATGGGTAAGGAGGACAAGAC-3′) and EF2T (5′- GGAAGTACCAGTGATCATGTT-3′) and the following PCR conditions: an initial cycle of denaturation at 95 °C for 1 min 25 s, 25 cycles of denaturation at 95 °C for 35 s, annealing at 57 °C for 55 s, extension at 72 °C for 1 min, and a final extension at 72 °C for 10 min. Amplification reactions were conducted in volumes of 25 µL containing 0.8 µM of each primer (Sigma-Aldrich), 0.2 mM of each dNTP, 1× NH_4_ Reaction Buffer, 2 mM MgCl_2_, 0.75 U of Taq DNA polymerase (BIOTAQ, Bioline, London, UK), and 20 ng of genomic DNA.

The PCR products were subjected to gel electrophoresis in the presence of ethidium bromide and visualized under UV light. The PCR amplicons were purified with the UltraClean^®^ 15 DNA Purification Kit (MOBIO Labs, Carlsbad, CA, USA), and sequencing was conducted using an ABI 3730xl genetic analyzer by Stab Vida Ltd. (Caparica, Portugal) in both directions with the same primers. The sequences were processed and edited using the 4Peaks program (version 1.8) and compared with the GenBank and Fusarium-ID databases (GenBank accession numbers corresponding to our isolates are MW244879–MW245002).

### 2.5. Pathogenicity Tests

Koch’s postulates were followed with 21 isolates chosen at random. For each isolate, an agar plug with a diameter of 1 cm was removed from the edge of 7-day-old mycelia and aseptically placed into a wound (with the same diameter and a depth of 5 mm) on symptomless disinfected cloves and covered with Parafilm^®^ (Bemis Company Inc., WY, USA). Garlic cloves inoculated with sterile agar plugs served as controls. Five garlic cloves (cv. ‘Morado de Cuenca’) per isolate were tested and replicated. After inoculation, the garlic cloves were packaged in plastic trays with wet filter paper and incubated at 25 °C for 14 d. To fulfill Koch’s postulates, specimens were isolated on PDA plates in aseptic conditions after 7 days of incubation at 25 °C from inoculated garlic cloves showing disease symptoms. The tests were conducted twice. This same experiment was conducted with non-wounded garlic cloves. For all the inoculated fungi, the disease severity index (DSI) was assessed using the clove rot index scale, which ranges from 0 to 3 (0: no symptoms; 1: incipient surface spots of brown rot ≤ 2 mm in diameter; 2: advanced coalescing brown rot spots that penetrate the clove; 3: brown rot covers ≥ 75% of the clove surface).

### 2.6. Statistical Analysis

The incidence (average percentage of bulbs showing a specific symptom) and severity (average percentage of affected cloves showing a specific symptom per bulb) of different disorders were calculated for each garlic bulb.

The effects of the harvest season and the garlic cultivar were evaluated for each symptom using logistic regression analysis. The Wald statistic was used to determine the significance of the variables with *p* ≤ 0.05. The relative frequency of each isolated fungal genus was calculated for each symptom.

The disease severity index (DSI) results were analyzed using a one-way ANOVA model with the IBM SPSS Statistics program (version 21.0), and average data were compared using Tukey´s pairwise comparisons at *p* ≤ 0.05.

A correlation analysis between the DSI data and the production of mycotoxins (fumonisin B1 (FB_1_), fumonisin B2 (FB_2_), fumonisin B3 (FB_3_), beauvericin (BEA), and moniliform (MON)) was conducted in previous work [20], and the growth rates of 20 inoculated isolates were determined.

## 3. Results

### 3.1. Observed Symptoms on Garlic Cloves

Based on the binary logistic regression analysis (fungal isolation frequency over 80%), represented in Table 1, three symptoms were associated with fungal species according to external and internal characteristics, as shown in Figure 2.

The first symptom detected, named “Dry Rot” (DR), is characterized by the presence of brown spots with a dry appearance and variable size (from less than 1 mm to completely covering the garlic clove), an irregular shape, and/or internal progression. The presence of white mycelia can be observed. *Fusarium* was isolated in 76% of the samples analyzed with this symptom (Table 1). The second symptom, named “Green Rot” (GR), is characterized by the presence of depressed yellowish spots of aqueous appearance and undefined shape. Afterward, the characteristic green mycelia were observed. *Penicillum* was isolated in 78.1% of these symptomatic samples (Table 1). The last symptom, named “Wound Damage” (WD), manifests as spots with variable colors and is characterized by mycelial growth, a dry appearance, and a defined contour. No internal progression was observed in the clove flesh. This symptom was associated with four fungal genera: *Stemphylium*, *Fusarium*, *Embellisia*, and *Penicillium* (Table 1).

### 3.2. Incidence and Severity of Post-Harvest Fungal Diseases

A total of 261 bulbs and 1264 garlic cloves were analyzed. The DR symptom was most frequently observed in garlic bulbs (56.1%). The GR symptom was detected in 12.2% of the analyzed bulbs, and the WD symptom was present in 5.1% of the samples. On average, the DR symptom occurred on 20.9% of affected cloves per bulb. The severity percentages of GR and WD were 2.5% and 0.8%, respectively.

The binary logistic regression analysis performed the GR for incidence data showed significant differences between seasons (*p* ≤ 0.001) and garlic cultivars (Chinese White, *p* ≤ 0.001; Spanish White, *p* = 0.024) (Table 2). This symptom was detected more frequently in the White cultivars than in the Purple cultivars.

### 3.3. Identification of the Fungal Species Associated with Each Symptom

*Fusarium* spp. was the most frequently isolated genus (76%) from cloves with the dry rot symptom (Table 1). *Fusarium proliferatum* was the most predominant species (85.5%), followed by *F. oxysporum* (13.2%), *F. solani* (2.6%), *F. subglutinans* (1.3%), *F. redolens* (1.3%), and *F. acuminatum* (1.3%). The *F. proliferatum* colonies on PDA grew white to purple mycelia and catenate microconidia (club-shaped with a flattened base, aseptate) and produced mono- and polyphialides. The curved macroconidia usually had 3–5 septates, and chlamydospores were not detected. The *EF-1α* sequences (~710 bp) of the isolates, previously assigned to *F. proliferatum* according to morphological characteristics, shared 99% identity with *F. proliferatum* (AF291058) based on BLASTn analysis in the Fusarium-ID database. The *F. solani* colonies produced sparse white to cream mycelia and oval-ellipsoid microconidia with 0 or 1 septa grouped in false heads over long monophialides. Numerous macroconidia with 5–7 septa were relatively wide, straight, stout, and robust with blunt and rounded apical cells. Chlamydospores formed individually or in pairs. The partial *EF-1α* sequences (~700 bp) from morphologically identified isolates showed 99% identity with *F. solani* (JF740714). The *F. oxysporum* colonies grew white to pale-violet mycelia with oval-elliptical and zero-septate microconidia were produced in false heads over short, bottle-shaped monophialides. The fusiform macroconidia had tapered and curved apical cells and foot-shaped to pointed basal cells with 3–5 septa. Chlamydospores formed individually, in pairs, or in short chains. For most of the previously identified isolates, partial *EF-1α* sequences (~690 bp) showed 99% identity with *F. oxysporum* f. sp. *cepae* (KP964904), but one isolate shared closer identity (99%) with *F. redolens* (HQ731060). The *F. acuminatum* colonies produced rose mycelia and a carmine-red undersurface on PDA. Microconidia formation was not observed. The macroconidia had elongated basal cells, and 3–5 septa were formed by short phialides. Abundant macroconidia formed chains and clusters. The partial *EF-1α* sequences (~715 bp) for these isolates shared 99% identity with *F. acuminatum* (KC175292). The *F. subglutinans* colonies had abundant mycelial growth, which were initially white and became violet as the culture aged. Agar pigmentation ranged from colorless to dark purple. Slightly curved macroconidia with curved apical cells and relatively poorly developed basal cells were observed. Oval and zero-septate microconidia were formed by false heads in mono- and polyphialides. The partial *EF-1α* sequences (~650 bp) showed 99% identity with *F. subglutinans* (JF278583).

*Penicillium* was the most frequently isolated genus (78.1%) associated with green rot (Table 1). *Penicillium* isolates produce rough, strong terverticillate conidiophores and phialides with green globose to subglobose conidia. Sclerotia were not observed. The analysis of *TUB* sequences (~470 bp) showed 99% identity with *Penillium allii* (HQ695995). A total of 11 isolates of *P. allii* were incubated on Czapek yeast agar (CYA) medium, and three morphotypes were differentiated by the texture of the colony, the presence of exudates, diffusible yellow pigment, and concentric furrows (Table 3).

The colony diameter of all isolates ranged from 17.63 to 33.29 mm. Morphotype 1 was characterized by a granular appearance, no production of yellow exudates, low production of diffusible yellow pigment, and the absence of concentric furrows (Appendix A). Colonies of Morphotype 2 had a velvety colony texture, small drops of yellow exudate, high production of diffusible yellow pigment, and concentric furrows. Morphotype 3 was represented by six isolates and characterized by a granular colony texture, small drops of yellow exudate, low production of diffusible yellow pigment, and concentric furrows.

Finally, a single fungus associated with wound damage symptomatology was not isolated, and the different isolated fungi never exceeded 50% of the samples analyzed. *Stemphylium* (42.9%), *Fusarium* (28.6%), *Penicillium* (21.4%), and *Embellisia* (14%) were the four fungal genera isolated for this symptom. *Stemphylium vesicarium* colonies were cottony with a color ranging from grayish to black with a white halo and black underside. The conidiophores were cylindrical, unbranched, and light brown with dark brown apical swelling at the site of conidium production. Mature conidia were brown, oblong, or broadly oval, with 1–3 transverse constrictions, 1–3 complete or nearly complete series of longitudinal septa, and 1–5 transverse septa. The ITS sequences showed 99% identity with *Stemphylium vesicarium* (LN896693). The *Embellisia allii* colonies presented partially superficial and immersed brown-blackish mycelia. The conidia were brown, solitary, smooth, oblong, and ellipsoid to cylindrical with 4–5 transverse septa. The conidiophores were brown, smooth, curved, and septate in groups of 3–6 solitary conidia. The ITS sequences showed 99% identity with *Embellisia allii* (JN588614).

### 3.4. Postharvest Pathogenicity Test on Garlic Cloves

Garlic cloves were inoculated with the most representative isolated fungus species. A total of 28 isolates belonging to three *Fusarium* species were tested for their ability to cause rot in garlic cloves. Brown superficial lesions were observed on the non-wounded cloves inoculated with *Fusarium* spp. White mycelial growth and internal brown lesions were observed when wounded cloves were inoculated, showing the same characteristics as the original symptomatic cloves (Appendix A). *Fusarium* spp. caused garlic clove rot with different levels of severity in the inoculated cloves, indicated by a DSI ranging from 1 to 3 (Figure 3).

All isolates of *F. proliferatum* were pathogenic to garlic cloves. *Fusarium proliferatum* isolates showed high virulence when inoculated, and *F. oxysporum* and *F. solani* isolates were moderately virulent. The disease severity scores for *F. proliferatum* isolates inoculated on wounded cloves were 2.14 on average. *Fusarium oxysporum* and *F. solani* resulted in lower disease severity with means of 1.04 and 1.23, respectively. In contrast, all isolates of *F. proliferatum* were pathogens, and 11 out of 20 were highly virulent. Correlation analysis of 20 *F. proliferatum* isolates did not reveal any significant correlations between the virulence (DSI), mycotoxin production (FB_1_, FB_2_, FB_3_, BEA, and MON), and mycelial growth (mm/day) (Table 4).

*P. allii* isolates produced small, superficial lesions on non-wounded garlic cloves. In contrast, when wounded cloves were inoculated, green mycelial growth and internal progression of the lesion were observed. Inoculations *with P. allii* isolates reproduced the same symptoms as the original specimen. The DSI ranged from 2.17 to 3 (Figure 3). Four out of five isolates were considered highly virulent, and *Embellisia allii* and *S. vesicarium* were pathogenic to wounded garlic cloves. Superficial mycelial growth was observed on the wounded cloves, but no internal lesions were observed.

## 4. Discussion

Postharvest rot caused by fungi strongly affects the quality of harvested garlic bulbs and produces different alterations in cloves that devalue the final product. The most frequently observed fungal diseases on harvested bulbs in Spain were primarily caused by two fungal genera: *Fusarium* (isolated from 76% of symptomatic garlic) and *Penicillium* (isolated from 78.1% of symptomatic garlic). “Wound Damage” was detected in 5.1% of the samples; however, our results indicate that this symptom is caused by trauma or injuries that occur during harvest or postharvest handling, followed by the growth of pathogenic or weakly pathogenic fungi, such as *Fusarium*, *Penicillium*, *Embellisia* or *Stemphylium*.

The appearance of the disease caused by *Fusarium* spp. differs depending on its evolution or environmental conditions. In the initial phase, brown lesions with a dehydrated appearance may emerge in different zones of the garlic clove. These spotted lesions are a few millimeters and can progress to cover the entire clove during the storage period. This disease is caused by *Fusarium* spp., and it is consistent with previous works by different authors [14,15,28,29,30]. In some cases, these lesions are covered with white mycelia. Moreover, this disease can be confused with the damage caused by eriophyid mites which cause a superficially dehydrated appearance that subsequently darkens and can be confused with Fusarium dry rot. Cloves affected by physiological disorders, such as waxy breakdown, turn deep yellow/amber and subsequently darken later, and they are easily confused with Fusarium dry rot. Of the postharvest symptoms observed in this study, Fusarium dry rot was detected with the highest frequency in Spain. The high incidence is consistent with Moharam et al. [14], who reported that *Fusarium* spp. were the most frequently isolated pathogen from garlic bulbs in the field and during storage (detected in up to 66.7% of the samples analyzed) in Egypt. In Italy, Mondani et al. [15] observed that *F. proliferatum* was the predominant fungus in infected bulbs (mean incidence: 35.4%) and was confirmed to be the causal agent of rot in garlic in the postharvest period. Logistic regression analysis revealed that the harvest season and garlic cultivar had no significant effects on the incidence of this disease. Six species of *Fusarium* were detected on affected cloves: *F. proliferatum*, *F. oxysporum*, *F. solani*, *F. acuminatum*, *F. subglutinans*, and *F. redolens.* The most prevalent species was *F. proliferatum*, detected in more than 85% of symptomatic cloves, followed by *F. oxysporum* and *F. solani*. Several studies have previously cited *F. proliferatum* as the causal agent of garlic bulb rot in different countries such as Hungary [31], USA [29], Spain [32], Italy [30], Argentina [33], India [34], France [35], and Russia [36]. Histological observations of *F. proliferatum* during the early stages of infection have confirmed that it colonizes roots 72 h after inoculation, although it does not appear to directly penetrate healthy cells [5]. All the tested *F. proliferatum* isolates were pathogenic to garlic cloves, whereas some isolates of *F. oxysporum* and *F. solani* were not, consistent with Moharam et al. [14]. *Fusarium oxysporum* seems to play a role in infections on basal plates and roots [15]. Moreover, *F. proliferatum* was more virulent than *F. solani* and *F. oxysporum* for both non-wounded and wounded cloves. Variability in the virulence of *F. oxysporum* and *F. proliferatum* has also been cited for other species within the *Allium* genus, such as onion [37], leek, and chive [17].

*Penicillium allii* was previously described by Dugan [38], Valdez et al. [39], and Moharam et al. [14] as the causal of green rot disease in garlic in the United States, Argentina, and Egypt, respectively. *Penicillium allii* was detected in 12.2% of the samples analyzed, which is consistent with Valdez et al. [13] and Moharam et al. [14], who observed losses ranging from 8.25% to 18.18% (13.21% on average) of harvested garlic bulbs in Argentina and Egypt, respectively. The disease was more frequently observed in White cultivars (Chinese and Spanish) than in Purple cultivars, and this incidence increased during the 2014 season. Overy et al. [40,41] cited *P. allii* as the most virulent fungal pathogen affecting garlic plants in the field and during subsequent storage. The comparison of the *TUB* gene sequences of the Spanish isolates led to the identification of *P. allii*, which comprises different morphotypes that vary in the texture of the colonies, the presence of exudates, the production of diffusible yellow pigment, and the morphology of the observed furrows. Valdez et al. [13] also described several morphotypes among *P. allii.* All isolates showed high virulence on previously wounded garlic cloves [12,13,42]. Our results show that *P. allii* was less virulent on non-wounded cloves, indicating that fungal infections require the presence of previous wounds to develop [43]. The prevention of trauma and wounds during the harvesting and subsequent handling of garlic is fundamental for effectively controlling postharvest green rot.

*Stemphylium vesicarium* and *E. allii*, characterized by the presence of dark-colored mycelia, were isolated from cloves with the “Wound Damage” symptom. *Stemphylium vesicarium* has been described as the causal agent of leaf blight of garlic [44,45], but its isolation from postharvest garlic cloves is described in this paper for the first time.

*Embellisia allii* (syn. *Alternaria embellisia*) has been previously reported as causing cosmetic blemishes on the outer skin layers of the bulb [46,47,48]. Our results show the presence of this pathogen on postharvest bulbs and its pathogenicity to wounded cloves. These results agree with those recently cited as causing bulb canker in garlic in Mexico [49]. Inoculation tests with both fungi confirm that they cannot penetrate the clove and cause internal rot. However, *E. allii* was able to grow to some extent and sporulate on the wounded surfaces of cloves. These results suggest that these wounds were caused during cultivation or harvesting and at the postharvest stage, facilitating the development of these fungi but without progressing to clove rot. Therefore, these damages should not be underestimated, as they may not only serve as a possible source of the primary inoculum but also facilitate the long-distance dispersal of propagules.

Garlic is typically propagated via seed cloves, and ensuring that materials are pathogen-free is very important for preventing the further development of diseases. Meristem culture is commonly used by growers to obtain virus-free cloves. Planting aerial bulbils might be an alternative approach to propagation with less infection by pathogens than that observed with seed cloves [50]. Fungicides and biological control might diminish the infection of fungal pathogens on postharvest bulbs [12,51,52], but the safety periods of different active ingredients, together with long postharvest periods, considerably reduce their effectiveness. On the contrary, the effect of gaseous ozone treatment during postharvest storage was recently evaluated, and the results for postharvest rot are promising [53]. These and other tools should be aimed at reducing the inoculum density to avoid the progression of diseases such as dry and green rot in garlic in Spain, the first garlic producer in the EU.

## 5. Conclusions

*Fusarium proliferatum* and *Penicillium allii* were identified as the main causal agents of postharvest fungal diseases in Spanish garlic. The most prevalent postharvest disease was Fusarium dry rot, which was primarily associated with *F. proliferatum.* To the author’s knowledge, this is the first report of *Fusarium solani* causing Fusarium rot of garlic during storage in Spain.

*Stemphylium vesicarium* and *Embellisia allii* were also detected in the postharvest stage. The presence of these species is relevant because they are garlic pathogens and affect plants during cropping, especially *S. vesicarium*, the causal agent of leaf blight of garlic, and the affected garlic cloves can serve as inoculum reservoirs. Our results provide useful insights into fungal diseases in garlic bulbs and suggest mechanisms for the early detection of symptoms during the storage period. Such strategies should be implemented when possible and may be useful in preventing the progression of the disease.

## 6. Patents

This section is not mandatory but may be added if there are patents resulting from the work reported in this manuscript.

## Figures and Tables

**Figure 1 foods-10-01063-f001:**
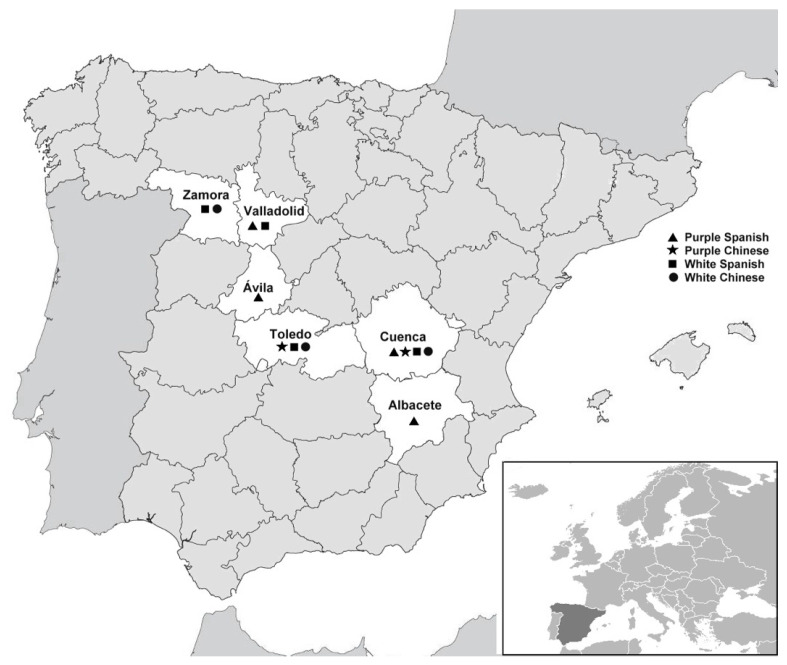
Map of Spain indicating the six provinces surveyed in this study (white area). The different symbols indicate the type of garlic analyzed in each province.

**Figure 2 foods-10-01063-f002:**
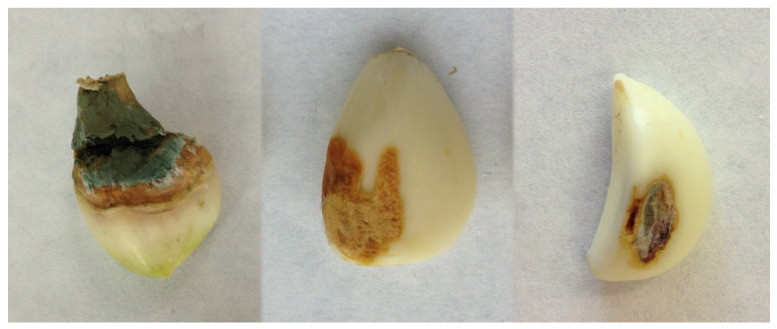
Symptoms associated with fungal species isolated from garlic cloves collected in Spain. Dry rot (**left**), green rot (**middle**), and wounded damage (**right**).

**Figure 3 foods-10-01063-f003:**
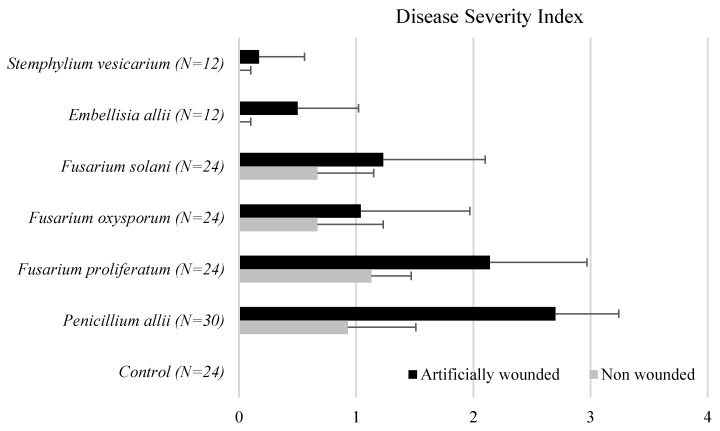
Disease severity index (DSI) on garlic cloves non-wounded and artificially wounded inoculated with all fungal isolates. Data (mean ± SD) for each clove garlic were graded into four classes (0–3) from no symptoms to severe rot. The same letter indicates no significant difference at *p* ≤ 0.05 according to ANOVA tests.

**Table 1 foods-10-01063-t001:** Percentages of fungal genera isolated from symptomatic garlic cloves on the samples collected in two consecutive harvest seasons.

Symptom	*n*	Fungal Prevalence	*Aspergillus*	*Embellisia*	*Fusarium*	*Penicillium*	*Rhizopus*	*Stemphylium*
Dry Rot	104	80.8	0.0	0.0	76	5.8	3.8	0
Green Rot	32	81.3	3.1	0.0	3.1	78.1	0.0	0
Wound Damage	14	100.0	0.0	14	28.6	21.4	0.0	42.9

**Table 2 foods-10-01063-t002:** Percentages of symptomatic cloves (incidence means) depending on the harvest season and cultivar according to logistic regression analysis.

Symptom	Season		Garlic Cultivar			
	2013	2014	Spanish Purple	Spanish White	Chinese Purple	Chinese White
GR ^1^	90.3 *	9.7 *	9.7	35.5 *	16.1	38.7 *
DR ^2^	40.1	59.9	21	24.9	29.3	24.8
WD ^3^	58.3	41.7	0	7.5	25	67.5

^1^ GR: Green Rot; ^2^ DR: Dry rot; ^3^ WD: Wound Damage. * Incidence data show significant differences (*p* ≤ 0.05).

**Table 3 foods-10-01063-t003:** Morphological characteristics of the fungal colonies of *P. allii* on CYA medium after 7 days at 25 °C and their classification into different groups. The mean diameter of the mycelial colony ± standard deviation is represented.

Isolate	Colony Diameter (mm)	Texture	Exudate	Diffusible Yellow Pigment	Concentric Furrows	Group
PA086	33.29 ± 1.21	Granular	−	Low	Absent	1
PA088	17.63 ± 0.99	Velvety	+	High	Present	2
PA089	31.97 ± 0.79	Granular	+	Low	Present	3
PA090	32.74 ± 0.93	Granular	−	Low	Absent	1
PA100	27.55 ± 0.50	Velvety	−	Low	Present	2
PA102	30.54 ± 0.64	Velvety	+	High	Present	3
PA112	27.72 ± 0.97	Velvety	+	Low	Present	3
PA113	27.46 ± 0.34	Velvety	+	Low	Present	3
PA114	30.83 ± 0.83	Granular	+	High	Present	3
PA115	30.57 ± 0.46	Granular	−	High	Present	3
PA130	21.75 ± 0.59	Granular	+	Low	Absent	1

**Table 4 foods-10-01063-t004:** Description of the *F. proliferatum* isolates indicating their growth rates (mm/day), DSI values of inoculated garlic cloves and levels (in µg/g) of *fumonisins* (FB_1_, FB_2_, and FB_3_), *beauvericin* (BEA), and *moniliformin* (MON) produced on rice grain substrate in vitro.

Isolate	FB1	FB2	FB3	BEA	MON	Growth Rate	DSI
FPG05	749.59	144.82	45.05	146.09	57.62	8.34	1.42
FPG08	2100.2	389.16	72.75	412.25	0.27	10.13	1.50
FPG12	55.55	0.73	0.57	nd	0.98	5.09	2.17
FPG16	8.64	1.36	0.43	188.48	11.59	8.09	1.00
FPG20	766.74	127.79	13.58	429.78	0.44	9.21	2.09
FPG21	2913.99	708.54	104.08	472.02	0.24	8.17	1.92
FPG23	120.06	25.56	19.37	152.13	0.13	12.04	2.25
FPG30	527.03	88.54	43.51	27.56	2.38	11.67	2.17
FPG33	667.83	158.36	45.1	995.37	nd	7.80	2.25
FPG34	1202.61	34.11	19.92	44.37	68.88	9.13	1.17
FPG35	239.29	22.4	18.27	79.07	1.26	11.79	2.25
FPG36	64.22	8.84	0.93	nd	4.71	9.13	0.92
FPG45	493.51	62.28	47.07	187.25	0.37	12.29	2.21
FPG58	1335.2	222.53	40.98	552.51	0.3	8.04	1.92
FPG59	249.06	40.36	5.86	68.65	5.71	7.59	1.17
FPG65	510.22	172.64	19.41	564.22	1.07	10.50	1.59
FPG75	1.64	0.31	0.2	96.24	0.29	7.21	1.00
FPG77	354.32	72.74	9.04	358.51	1.01	6.00	1.83
FPG80	586.94	86.15	14.67	160.08	8.14	6.67	2.25
FPG82	1730.23	216.28	15.96	596.94	1.87	9.84	1.42
*p* valor *	0.804	0.577	0.252	0.242	0.158	0.567	

nd: not detected; * bivariate correlations with virulence represented by DSI.

## Data Availability

Not applicable.

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
