# Peer review of "Incidence and Etiology of Postharvest Fungal Diseases Associated with Bulb Rot in Garlic (Alllium sativum) in Spain"

_foods, 2021, doi:10.3390/foods10051063_

Round 1

Reviewer 1 Report

Minor comments:

  • Please review the entire text for the correct use of abbreviations.

Author Response

Thanks for the feedback, we have done a thorough review of the manuscript and found some spelling errors, but we have not detected any wrong use of abbreviations. I do not know if the reviewer could indicate an example in the text to be able to modify it.

Reviewer 2 Report

the requested changes have been made, so I have no further recommendations 

Author Response

Comments and Suggestions for Authors: "the requested changes have been made, so I have no further recommendations". Response: Thank you for your previous recommendations. They have greatly improved the article.

This manuscript is a resubmission of an earlier submission. The following is a list of the peer review reports and author responses from that submission.

Round 1

Reviewer 1 Report

The study is original, the purpose and the aim of the work are described in an appropriate manner.

The methods, the reagents and the analysis are described with sufficient details, also through references to previous work.

The results are clearly exposed and the conclusions are well justified and supported by the obtained findings.

However there are several errors that could have been avoided with a more appropriate check.

Minor comments:

Introduction:

  • Lines 27-35: Please, remove this sentence.
  • Lines 42 and 54: Add references.
  • It would be interesting to mention in the Introduction the production of mycotoxins by fungal species.

Results:

  • In table 1 there are some abbreviations which are defined later in the text. Please review the entire text for the correct use of abbreviations.
  • Lines 150 and 163: DSI means “disease severity index” or “severity data”?
  • Lines 238-240: There are two tables named as “Table 2”. Please check.

Discussion:

  • Lines 340-343 and 345-347: Try to combine the two sentences to avoid repetition.

Other:

  • Line 393-419: Please check.

Author Response

Thank you so much for your comments, following we provide you a point-by-point response.

The study is original, the purpose and the aim of the work are described in an appropriate manner.

The methods, the reagents and the analysis are described with sufficient details, also through references to previous work.

The results are clearly exposed and the conclusions are well justified and supported by the obtained findings.

However there are several errors that could have been avoided with a more appropriate check.

Minor comments:

Introduction:

Lines 27-35: Please, remove this sentence.

Answer: Selected lines have been removed

Lines 42 and 54: Add references.

Answer: References 4 and 5 have been included in both sentences.

It would be interesting to mention in the Introduction the production of mycotoxins by fungal species.

Answer: The suggestion has been addressed and the mention on the mycotoxins production has been included (lines 96 - 102) furthermore, some new references has been included.

Results:

  • In table 1 there are some abbreviations which are defined later in the text. Please review the entire text for the correct use of abbreviations.

Answer: Yes, we realized that these abbreviations were defined later so we decided to include the full name for symptoms in table 1 (line 216) .

Lines 150 and 163: DSI means “disease severity index” or “severity data”?

Answer: We have checked all the document and rewrite when necessary (line 203)

Lines 238-240: There are two tables named as “Table 2”. Please check.

Answer: Absolutely, table 3 has been renumbered. (lines 289 and 290)

Discussion:

  • Lines 340-343 and 345-347: Try to combine the two sentences to avoid repetition.

Answer: Lines have been combined to avoid repetition (Lines 411 to 413)

Other:

  • Line 393-419: Please check.

Answer: We have verified that the pdf that has reached the reviewers does not match the Word document that we have uploaded to the platform.  In some formal aspects such as italics, but the authorship information does not appear as well, and paragraph with rules appears in the introduction which are not in the original either. We assume that it was an error when constructing the pdf with the data uploaded to the platform, but we thank the comment and we will contact the editor to find the best way to solve it (maybe upload again those data) and that it will appear correctly in the final version.

Reviewer 2 Report

This work on incidence and etiology of postharvest fungal diseases associated with rot bulb on garlic (Alllium sativum) in Spain is very interesting, especially for Spain as a major European producer. In this regard, some minor revisions are suggested:

  • Introduction: delete from line 27 to 34.
  • Line 37: social perspective, in which sense? I don’t understand. So clarify your statement, please.
  • Line 112: correct “specie level”
  • Figure 2. I suggest to present images following the description order of the text. The first symptom detected so it’s DR, and then GR. Exchange the first two images, please.
  • Paragraph 3.3. I suggest to italicize the scientific name of toxins, both in this paragraph and in the whole text.
  • Figure 3 caption: put non-wounded and artificially wounded before the subject garlic cloves.
  • Please complete Supplementary Materials with the captions of Figure S1 and S2.
  • Author Contributions: complete the paragraph with the initials of the authors
  • Funding: select the right sentence
  • Data Availability Statement: select the right sentence
  • Conflicts of Interest: select the right sentence
  • References: revise all references following the guidelines of the journal. They are formatted unevenly and there is a double numbering.
  • You didn’t mention figure S2 in the text.

Author Response

Thank you for your comments, following we provide a point by point response.

This work on incidence and etiology of postharvest fungal diseases associated with rot bulb on garlic (Alllium sativum) in Spain is very interesting, especially for Spain as a major European producer. In this regard, some minor revisions are suggested:

  • Introduction: delete from line 27 to 34.

Answer: Indeed, as the reviewer indicates, it seems that when the pdf was built, part of the instructions had been included, but it did not appear in the original document in Word uploaded by us, it was automatically included when the pdf was built. As we have indicated to the first reviewer, we thank the comment and we will contact the editor to find the best way to solve it in the final version.

.Line 37: social perspective, in which sense? I don’t understand. So clarify your statement, please.

  • Answer: It refers to a very important aspect in our country since garlic crop is grown in rural areas where it is one of the few crops with sufficient added value to allow the stabilization of the population in those rural areas, in any case this aspect has been deleted upon request from another reviewer.
  • Line 112: correct “specie level”.

Answer: The suggestion has been addressed

  • Figure 2. I suggest to present images following the description order of the text. The first symptom detected so it’s DR, and then GR. Exchange the first two images, please.

Answer: The suggestion has been addressed (line 146)

  • Paragraph 3.3. I suggest to italicize the scientific name of toxins, both in this paragraph and in the whole text.

Answer: The suggestion has been addressed

  • Figure 3 caption: put non-wounded and artificially wounded before the subject garlic cloves.

Answer: The suggestion has been addressed (lines 325-326)

  • Please complete Supplementary Materials with the captions of Figure S1 and S2.
  • Answer: The suggestion has been addressed (lines 680-682 and 689-691)
  • Author Contributions: complete the paragraph with the initials of the authors; Funding: select the right sentence; Data Availability Statement: select the right sentence; Conflicts of Interest: select the right sentence
  • Answer: Again, this information was uploaded in the platform and appear in the website, but it seems it has not been recorded; we will check with the editor.
  • References: revise all references following the guidelines of the journal. They are formatted unevenly and there is a double numbering.

Answer: All references have been revised

  • You didn’t mention figure S2 in the text.

Answer: It is mentioned in line 323 of the Word document, section 3.4.

Reviewer 3 Report

The article is written in plain, easy language. The manuscript requires some corrections.

There is no specific information in the Abstract. This should be redrafted by adding relevant conclusions from the work.
Authors should increase the description and characteristics of garlic as a plant. Please use the article:
(2020). Biological Activity of Some Aromatic Plants and Their Metabolites, with an Emphasis on Health-Promoting Properties. Molecules, 25 (11), 2478. doi:10.3390/molecules25112478
This will make it easier for the authors to enhance the characteristics of garlic.

Throughout the manuscript, the names of microorganisms are written in italics. This is an unacceptable error

The authors should add photos of agarose gels from the isolation of DNA of microorganisms. Authors should perform a statistical analysis of all the results in each of the tables and figures. It is very important. The results should also be statistically described in the discussion.

Garlic is propagated mainly vegetatively. Unfortunately, the used spreading material is not always of good quality. It happens that it is the source of many pathogens, which is a common cause of this vegetable's disease. The authors should describe examples of preventing the development of garlic diseases. This issue should be greatly expanded with new information.

References are poorly prepared. This needs to be corrected (Authors should read the requirements for authors).

The authors used a lot of old literature. This should be changed by adding the newest.

What is the industrial use of the obtained results. This should be described at work.

Author Response

Thank you so much for your comments, following we provide you a point-by-point response

The article is written in plain, easy language. The manuscript requires some corrections.

There is no specific information in the Abstract. This should be redrafted by adding relevant conclusions from the work.

.Answer: Comments are welcome, although the abstract should be a total of approximately 200 words maximum, we have added additional information within the abstract to better emphasize relevant results.

Authors should increase the description and characteristics of garlic as a plant. Please use the article:
(2020). Biological Activity of Some Aromatic Plants and Their Metabolites, with an Emphasis on Health-Promoting Properties. Molecules, 25 (11), 2478. doi:10.3390/molecules25112478
This will make it easier for the authors to enhance the characteristics of garlic.

Answer: We have reviewed the suggested article but it does not include any reference to garlic, neither to any of the substances such as allicins or alliins. In any case we understand the meaning of the comment and therefore we have completed the description of the characteristics of garlic (lines 38 to 43) and adding the following reference:

Ansary J, Forbes-Hernández TY, Gil E, Cianciosi D, Zhang J, Elexpuru-Zabaleta M, Simal-Gandara J, Giampieri F, Battino M. Potential Health Benefit of Garlic Based on Human Intervention Studies: A Brief Overview. Antioxidants (Basel). 2020 Jul 15;9(7):619. doi: 10.3390/antiox9070619.

Throughout the manuscript, the names of microorganisms are written in italics. This is an unacceptable error !!!! The authors make such mistakes that it is a shame !!!

Answer: We have read the reviewer's comment with surprise, since all the species have been written in italics (as can be seen in the original document sent by us, the document in .doc format). Therefore, I want to make it clear that in the work sent to the journal all the microorganisms were in italics.

We understand that it has been a problem when generating the pdf that is sent to the reviewers, as we have detected many other errors in the pdf such as not including the contribution of the authors or including the writing rules before the original text in the introduction. We will contact the editor to find the best way to solve it in the final version

The authors should add photos of agarose gels from the isolation of DNA of microorganisms. Authors should perform a statistical analysis of all the results in each of the tables and figures. It is very important. The results should also be statistically described in the discussion.

Answer: We understand that once the sequences of the different regions have been deposited in the open databases, it is not necessary to lengthen the article with photographs of gels that do not provide any additional information. Regarding the statistical analysis, we have reviewed the tables and figures, completing the type of analysis carried out when necessary. For example, the titles of Table 2 and Figure 3 have been supplemented with more information on the statistical analysis for clarity. Furthermore, in Figure 3 the statistically significant differences from the ANOVA analysis performed have been added.

Garlic is propagated mainly vegetatively. Unfortunately, the used spreading material is not always of good quality. It happens that it is the source of many pathogens, which is a common cause of this vegetable's disease. The authors should describe examples of preventing the development of garlic diseases. This issue should be greatly expanded with new information.

Answer: We strongly agree with the reviewer's comment, in our experience with the garlic crop, the health of the planting plant material is a key aspect. We have added information about it and some other aspects such as mycotoxins in the article. (lines 57-65; 96-102)

References are poorly prepared. This needs to be corrected (Authors should read the requirements for authors).

Answer: References has been revised and updated

What is the industrial use of the obtained results. This should be described at work.

Answer: In our experience, work is very necessary and demanded from the industry, there are several cooperatives and producers that have asked us for a job that allows them to identify these types of symptoms.

In the garlic dryers or in the warehouses where the bulb selection is carried out, early detection of garlic damaged by fungi is essential to be able to remove garlic showing disease symptoms from the food chain. This information has been included. I really want to thank the reviewer for his/her work to improve the article.

Round 2

Reviewer 3 Report

The authors did not correct the manuscript at all. There are no changes to the manuscript.

The authors did not even prepare an official version of the article in the FOODS template.